# Unexpected Basal Anti-Müllerian Hormone Concentrations in a 6-Year-Old Bitch Presenting an Ovarian Remnant

**DOI:** 10.3390/ani15030311

**Published:** 2025-01-23

**Authors:** Matteo Burgio, Lluis Ferré-Dolcet, Alice Carbonari, Lorenza Frattina, Annalisa Rizzo, Vincenzo Cicirelli

**Affiliations:** 1Department of Veterinary Medicine, University of Bari Aldo Moro, 70121 Bari, Italy; matteo.burgio@uniba.it (M.B.); lorenza.frattina@uniba.it (L.F.); annalisa.rizzo@uniba.it (A.R.); 2Independent Researcher, 08000 Barcelona, Spain; lluisferredolcet@gmail.com

**Keywords:** anti-müllerian hormone, bitch, ovariectomy, sex hormones

## Abstract

Anti-Müllerian hormone (AMH) is a hormone that plays a significant role in reproductive function. It is commonly used as a marker for ovarian reserve (the number of eggs remaining in a female) in humans and is also applied in veterinary medicine to assess reproductive status in dogs. AMH testing can help determine whether a female dog has been spayed, as spaying involves the removal of the ovaries, resulting in extremely low or undetectable AMH levels. Callie, a 6-year-old spayed female German Shepherd, was referred for suspected ovarian remnant syndrome (ORS). Her AMH concentrations were found to be extremely low, leading to a negative diagnosis for ORS based on bibliographic cut-off values. However, advanced diagnostic examinations subsequently confirmed the presence of ovarian tissue. This case highlights the importance of utilizing additional diagnostic methods to confirm ORS, as reliance on AMH testing alone may lead to false-negative results.

## 1. Introduction

The ovarian remnant syndrome (ORS) regroups all the affections in a spayed female presenting signs of an intact female. In addition, ORS is considered a complication during ovariectomy (OE) or ovariohysterectomy (OHE) with the failure of removing complete ovarian tissue in a bitch or queen [1,2,3].

Typical clinical signs of ORS include estrus signs such as vulvar swelling and discharge, behavioral changes with male attraction, mammary hyperplasia and lactation [2,3] with possible complications such as bilateral alopecia, pyometra or stump pyometra, mammary masses, and bone marrow aplasia due to steroid hormone impregnation [3,4,5,6,7].

Estrogen activity can be assessed by visualizing keratinized cells in a vaginal smear (18–20).

Diagnosis of ORS is performed by the identification of steroid hormones in the blood circulation after neutering or by the localization of a part or all of the ovary in the abdominal cavity. These procedures can be performed by exploratory laparotomy/laparoscopy [2,3,6,8,9], ultrasonography [10], and advanced imaging techniques [11]. In veterinary medicine, as showed in this case, abdominal ultrasound might not be sufficient for identifying adnexal structures due to their size, while computed tomography (CT) improves visualization and identification of ovarian tissue and helps avoid continuous exploration or subsequent surgical errors [12].

The presence of gonadal steroid hormones can also be used as a tool to identify ORS. Nevertheless, in some cases, small remnants of ovarian tissue might not produce sufficient estradiol levels capable of triggering a feedback mechanism of pituitary gonadotropins, resulting in aberrant or higher gonadotropin concentrations compared to intact bitches [13]. In fact, the pituitary gonadotropin LH has been considered a good biomarker to identify ORS, because it increases up to 30-fold in neutered animals due to the missing negative feedback of gonadal steroids. Basal concentrations of LH are found in intact females (excluding LH peak before ovulation) [14,15,16], although it should be considered that some reports have described a reduction to normal levels of LH in both neutered bitches and queens after some weeks of ovariectomy [10].

Progesterone (P4) is another valuable hormone for confirming ovarian remnant syndrome (ORS). Ovarian remnants may exhibit normal cyclicity, including the diestrus phase. Therefore, detecting high P4 concentrations in a neutered female dog could indicate the presence of corpora lutea. However, in this report, P4, LH, and estradiol concentrations were not measured during hematological analysis, which is an important limitation of this study.

To date, anti-müllerian hormone (AMH) detection has been described to be the gold standard technique for determining intact/neutered status in both queens and bitches [13,14,15,16]. AMH is a glycoprotein primarily produced by the granulosa cells in the ovaries in females and Sertoli cells in males [17,18]. It plays a vital role in sexual differentiation during embryonic development by inhibiting the development of Müllerian ducts, which would otherwise form female reproductive structures, such as the uterus and salpinges [19]. In fact, during fetal life, AMH levels are minimal but increase after birth, reflecting the ovarian follicle reserve and its development [10].

In adult subjects, AMH plays a significant role as a marker of gonadal function in both male and female animals. Its diagnostic value stems from its close relationship with the gonads, where it is produced, and its reflection of reproductive organ status and functionality [20,21,22]. In females, AMH is secreted by the granulosa cells in pre-antral and small antral follicles within the ovaries [10]. In fact, AMH levels give insight into this reserve, which decreases with age or after spaying. Therefore, measuring AMH is valuable for assessing fertility potential and diagnosing reproductive health issues [20].

In spayed females, AMH testing is crucial for the diagnosis of ORS. If ovarian tissue remains after surgery, AMH will still be detectable. Thus, elevated AMH levels in a spayed female would indicate that some ovarian tissue persists, warranting further intervention.

## 2. Case History

Callie, a 6-year-old, 29 kg spayed female mixed-breed German Shepherd, was brought to the Obstetrics and Gynecology Section of the Veterinary Teaching Hospital at Aldo Moro University for a specialist examination due to a serosanguineous discharge from the vulva and attraction to male dogs. The owner reported that Callie had undergone a partial ovariohysterectomy about 10 months prior to the visit to halt her reproductive activity, and after a few weeks, the vulvar discharge started. During the clinical examination, Callie had a body condition score (BCS) of 3/5, with a respiratory rate of 20 breaths per minute, heart rate of 90 beats per minute, temperature of 38.6 °C, and blood pressure of 110/80/90, all within normal limits. Abdominal palpation revealed no signs of pain. Upon examination of the genital area, the vulva appeared with normal dimensions with an evident serosanguineous discharge. A deep vaginal swab was collected and stained using Diff-Quik stain (Dif-stain, Titolchimica S.r.l., Pontecchio Polesine, Rovigo, Italy). The cytological image showed the presence of red blood cells without keratinized cells. The use of human compounds containing estradiol-derivates, such as pills or creams, was discarded by the owner. During the clinical examination, blood samples were taken for the evaluation of the main hematological, cytometric, and biochemical parameters, as well as for the evaluation of AMH concentrations to exclude the possibility of ORS. The results of the hematological and biochemical examinations showed no values that deviated highly from the normal reference ranges (Table 1 and Table 2).

Anti-Mullerian hormone analysis was performed using a chemiluminescence immunoassay on a cobas E602 analyzer (Roche) [23]. The results showed a serum concentration of 0.04 ng/mL, suggesting the absence of ovarian tissue, in accordance with previously described data [24,25,26,27,28,29]. To exclude the presence of space-occupying masses, foreign bodies, or other anatomical-functional alterations, abdominal ultrasound (US) and vaginoscopy were carried out.

The abdominal US showed the presence of slight reactivity and a cone of shadow at the level of the cervical stump remaining from the ovarian-hysterectomy surgery. The operator recommended a CT scan for a definitive diagnosis. On the other side, vaginoscopy showed no structural abnormalities or visible foreign bodies. Considering the negative results for the confirmation of an ORS, a computed tomography (CT) scan was carried out. A CT scan revealed a parenchymatous mass with slight, heterogeneous swelling, corresponding in shape and location to residual left ovarian tissue (Figure 1). Additionally, swelling of the uterus and vagina was detected being suggestive of estrus.

Following confirmation, the dog underwent an exploratory laparoscopy to remove the suspected ovarian remnant, which was subsequently preserved in 4% formalin at room temperature for 48 h until histological examination.

The excised structure was analyzed, and histological examination revealed that, in the examined sections, there were remnants of ovarian cortical, including degenerated corpora lutea and fragments of ovarian medulla. A morphologically normal portion of the uterine tube was evident (Figure 2).

## 3. Discussion

In the present case report, an adult neutered bitch presented sera-hemorrhagic vaginal discharge and attraction to male dogs, signs that began 10 months prior to presentation.

Signs of estrus in a neutered female dog prompt a differential diagnosis of ORS. The potential causes of ORS include surgical errors during the ovariectomy, revascularization of the residual ovarian tissue, auto transplantation of ovarian fragments left in the abdominal cavity, or the presence of a supernumerary ovary [4,5,6,7,30]. Other conditions that can mimic ORS include metastasis of granulosa cell tumor, adrenocortical tumors, intoxication with anticoagulants, and iatrogenic causes such as accidental ingestion or administration of hormones [4,13,14,15,16,31,32]. Diagnosis of ORS can be established by detecting reproductive steroid hormones in the blood or localizing ovarian tissue in the abdominal cavity through exploratory surgery [2,3,6,8,9], ultrasonography [10], or advanced imaging techniques like CT, which offers superior accuracy [11,12,33]. Hormonal analysis can aid in diagnosis, with elevated luteinizing hormone (LH) being a key indicator due to the lack of feedback from gonadal steroids in neutered animals. Progesterone (P4) levels may also indicate ORS when ovarian remnants retain cyclic activity. However, limitations such as insufficient estradiol production from small ovarian remnants or fluctuating hormone levels can make diagnosis challenging [14,15,16].

In the reported case, histological examination confirmed aging corpora lutea, suggesting that functional ovarian tissue was present. This finding is significant in the context of this case, given the time lapse of 10 months between the initial presentation and the exploratory laparoscopy. It is possible that the bitch was entering estrus during the first consultation and subsequently ovulated prior to the laparoscopy. This aspect is far from the physiological discharge of estrus, which commonly lasts for about 2 weeks. Furthermore, the bitch did not show all the signs of estrus; specifically, her vulva was not edematous. If this were the case, levels of these hormones at the initial consultation would have been at basal levels, rendering this test inconclusive. Unfortunately, this hypothesis cannot be confirmed.

While CT improved diagnostic accuracy compared to ultrasound, the absence of hormonal measurements (P4, LH, and estradiol) during the study represented a significant limitation, preventing confirmation of hormonal fluctuations associated with estrus or diestrus. To date, serum AMH identification is currently regarded as the gold standard for distinguishing between intact and neutered individuals, as AMH is produced by both Sertoli in males and granulosa cells in females. While scientific literature strongly supports the use of AMH for a definitive diagnosis, in this case, the results were inconclusive. Measuring AMH concentrations in dogs is typically performed using one of two primary methods, namely Enzyme-Linked Immunosorbent Assay (ELISA) or Chemiluminescent Immunoassay (CLIA). The literature reports varying AMH cutoff values for diagnostic purposes, influenced by factors such as the analytical method employed, the dog’s age, and the presence or absence of ovarian tissue. Themmen et al. (2016) [29], using the canine-based ELISA method, identified a cutoff of 1.1 ng/mL, achieving 100% sensitivity and 90% specificity for differentiating between intact and spayed females. In contrast, Walter et al. (2020) [34], employing the CLIA method, established a cutoff value of 0.06 ng/mL, achieving 100% sensitivity and specificity within a 95% confidence interval for the same purpose. In the current study, the patient presented an AMH level of 0.04 ng/mL, which fell below the cutoff values reported in the literature, complicating the diagnosis. Additional clinical signs (such as Reactive Prot. C and slightly elevated neutrophils counts) left the clinician at a diagnostic impasse, necessitating the use of advanced and costly diagnostic tools, including computed tomography, laparoscopy, and histology, to confirm the diagnosis of ORS.

While effective in most cases, current AMH cutoff values are not definitive and may lead to diagnostic uncertainty. Thus, complementary diagnostic tools such as vaginal smear, P4 assay, and advanced imaging techniques should be always considered in the diagnosis of ORS.

## 4. Conclusions

In conclusion, the present case report highlights the limitations of relying solely on AMH cutoff values and underscores the importance of integrating AMH testing with complementary diagnostic methods to achieve accurate and reliable outcomes. AMH levels near the established thresholds should be considered inconclusive, prompting further investigation using advanced diagnostic techniques with advanced imaging diagnostics.

## Figures and Tables

**Figure 1 animals-15-00311-f001:**
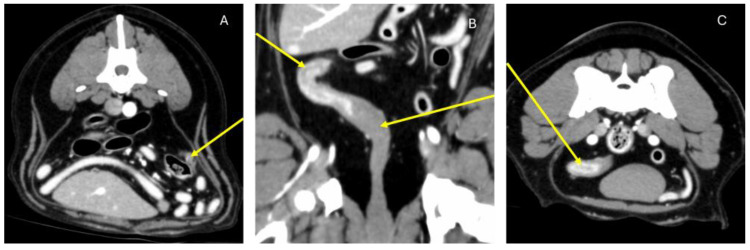
The arrows indicate the following structures: (**A**) abdominal structure compatible with residual ovarian structure; (**B**,**C**) uterine and cervicale structure remaining from OHE.

**Figure 2 animals-15-00311-f002:**
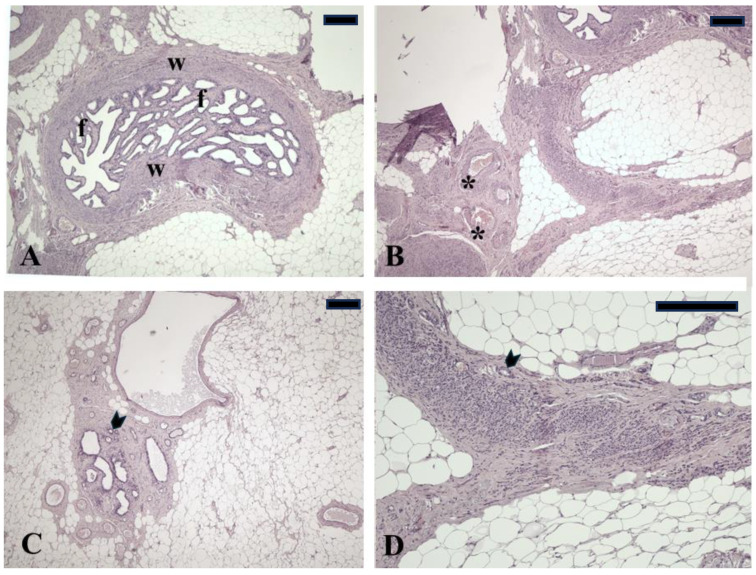
Micrographs from sections of residual oviduct and ovary fragments. (**A**) Oviduct showing extensive fibrosis affecting the wall (w) and the mucosal folds (f). (**B**) Ovary tissue showing cystic follicles (asterisks). (**C**) Ovary tissue showing few primordial follicles (arrowhead) in the cortical region. (**D**) Ovarian parenchyma showing loss of normal physiological architecture due to degenerative processes and few primordial follicles (arrowhead). Hematoxylin-eosin staining. Magnification bars = 100 µm.

**Table 1 animals-15-00311-t001:** Hematological examination.

Parameters	Sample	Reference Range	Parameters	Sample	Reference Range
RBC (milioni/µL)	8.43	5.50–8.00	WBC (×1000/µL)	14.2	6.0–14.0
HGB (g/dL)	20.5	14.0–19.5	Correct counting WBC (×1000/µL)	14.2	6.0–14
HCT (%)	60.9	38.0–54.0	Myelocytes (/µL)	0	0
MCV (fL)	72.2	60.0–73.0	Metamyelocytes (/µL)	0	0
MCH (pg)	24.3	21.0–27.0	Band neutrophils (/µL)	284	0–300
MCHC (g/dL)	33.7	33.0–37.0	Segmented neutrophils (/µL)	11,360	3500–9300
CHCM (g/dL)	33.2	32.0–37.0	Limphocytes (/µL)	1562	1200–3200
CH (pg)	23.9	21.5–25.7	Monocytes (/µL)	710	200–1200
CHDW (pg)	2.57	3.18–3.80	Eosinophils (/µL)	284	100–800
RDW (%)	13.2	12.0–16.0	Basophils (/µL)	0	0–100
HDW (g/dL)	2.33	1.50–2.50	Emoparasites	NEG.	NEG.
NRBC/100 WBC	0	0			
PLT (1000/µL)	138	180–450			
MPV (fL)	11.2	8.5–14.5			
PCT (%)	0.15	0.20–0.50			
PDW (%)	64.9	53.0–70.0			
MPC (g/dL)	23.4	18.0–24.0			
MPM (pg)	2.41	1.5–2.45			
Large PLT (1000/µL)	42	7–50			

RBC—red blood cell; HGB—hemoglobin; HCT—hematocrit; MCV—mean corpuscular volume; MCHC—mean corpuscular hemoglobin concentration; CHCM—mean optical hemoglobin concentration; CH—cellular hemoglobin; CHDW—cellular hemoglobin distribution width; RDW—red blood cells distribution width; HDW—hemoglobin distribution width; NRBC—nucleated red blood cell; WBC—white blood cell; PLT—Platelet; MPV—mean platelet volume; PCT—Plateletcrit; PDW—platelet distribution width; MPC—mean platelet component; MPM—mean platelet mass.

**Table 2 animals-15-00311-t002:** Biochemical examination.

Parameters	Sample	Reference Range	Parameters	Sample	Reference Range
CPK (IU/L)	58	30–160	Calcium (mg/dL)	10.5	9.0–11.5
AST (IU/L)	35	18–50	Corrected Calcium (mg/dL)	10.8	9.0–11.5
ALT (IU/L)	177	20–75	Phosphorus (mg/dL)	4.4	2.8–4.7
ALP (IU/L)	44	20–160	Magnesium (mg/dL)	1.8	1.4–2.1
GGT (IU/L)	4	0.65–6.5	Sodium (mEq/L)	143	140–152
Cholinesterase (IU/L)	4617	3350–7250	Potassium (mEq/L)	4.8	4.0–5.3
Total Bilirubin (mg/dL)	0.21	0.10–0.30	Na/K	29.8	28.0–37.0
Total Protein (g/dL)	5.9	5.7–7.3	Chlorine (mEq/L)	112	105–115
Albumins (g/dL)	2.9	2.8–3.7	Corrected Chlorine (mEq/L)	114.3	109–115
Globulins (g/dL)	3	2.8–3.9	HCO-3 (mmol/L)	20.8	15.8–22.3
A/G:	0.97	0.70–1.30	Anionic Gap	15.0	18.5–29.0
Cholesterol (mg/dL)	231	130–330	Osmol. sier. mis. (mOsm)		292–310
Triglycerides (mg/dL)	79	30–130	Osmol. sier. calc. (mOsm)	279	277–297
Amylase (IU/L)	839	472–1036	Osmol. Gap		18–28
Lipase (IU/L)	68	77–585	Total Iron (µg/dL)	92	90–230
Urea (mg/dL)	26	15–50	UIBC (µg/dL)	247	180–350
Creatinine (mg/dL)	1.13	0.70–1.40	TIBC (µg/dL)	339	300–500
Glucosie (mg/dL)	103	70–130	Saturation (%)	27.1	20–60
Lipase DGGR (UI/L)	29	26–186	Reactive Prot. C (mg/dL)	2.82	0.01–0.45

CPK—creatine phosphokinase; AST—aspartate amino transferase; ALT—alanine transaminase; ALP—alkaline phosphatase; GGT—gamma glutamyl tranferase; A/G—albumin to protein ratio; UIBC—unsaturated iron binding capacity; TIBC—total iron binding capacity.

## Data Availability

The data that supports the findings are contained in the paper.

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
