# Peer review of "Unexpected Basal Anti-Müllerian Hormone Concentrations in a 6-Year-Old Bitch Presenting an Ovarian Remnant"

_animals, 2025, doi:10.3390/ani15030311_

Round 1
Reviewer 1 Report
Comments and Suggestions for Authors
General comments
This is a case report on an important topic, which is of interest to small animal vets. It is often indeed a problem if a bitch is suspected of ORS and has low AMH: vets are then in doubt of the diagnosis and need to know a way on how to proceed. Here CT scan is proposed but other solutions are also possible.
However the paper is not acceptable in its current form and should be rewritten completely. A few suggestions for improvement are written below and in the specific comments, but the paper should also be checked for English language and figures need profound editing. The timing of all events needs to be better described in a timeline of the history of the dog.
1. In the introduction it should be explained better how diagnosis of ORS can be performed. There are different blood tests and other tests which can be performed for diagnosis of ORS – references to these method can be found in Manuals of dog reproduction or in papers such as cited here : “Currently, diagnostic methods such as medical history, oestrous symptoms, vaginal cytology (if in the follicular phase), hormone analysis (oestrogen and progesterone concentrations after GnRH stimulation test) and exploratory laparotomy are used to establish a diagnosis in ORS suspected dogs”(ref https://onlinelibrary.wiley.com/doi/10.1111/rda.13451)
2. Do not use the word sexual hormones. It is preferable to refer to steroid hormones (E2 and P4) and gonadotropins (LH & FSH).
3. Indicate that just like when determining optimal timing of breeding or insemination we do not rely solely on progesterone, diagnosis of ORS cannot rely solely on AMH. It is the combination of all symptoms that leads us to the diagnosis.
The paper needs a thorough rereview before it can be accepted for publication.
Specific comments
Simple summary needs to be revised for language
Abstract needs to be revised too
line 27-30 Please remove one of the following sentences since they give the same information :
“Following advanced diagnostics and laparoscopy, histological examination confirmed the presence of ovarian tissue in the dog’s abdominal cavity. In this clinical case, in addition to hormonal tests, a diagnosis was made using CT scans and laparoscopy, confirming the presence of ovarian tissue in the dog’s abdominal cavity”
Line 37 : : why is ORS a non-pathological condition? Please explain better or remove non-pathological
Line 45 : please rephrase : it is not sexual hormones that induce keratinization (or better cornification) but estrogens . Progesterone is also “a sexual hormone” (not sure if this is proper terminology) but does not induce cornification in the vagina.
Line 72: What is meant by a partial ovariohysterectomy? It is also not clear how the timing of OVH relates to the serosanguinous blood loss. Was the blood loss ten months before the OVH and the OVH 10 months before the current visit (so blood loss 20 months ago?) , or did the blood loss and OVH occur at the same time (meaning that she should have had another estrous cycle about 4 to 2 months ago (6-8 months interoestrous interval in most dogs). Was the dog spayed during oestrus or 3 months after oestrus as adviced in textbooks? At what time of the cycle was the final curative surgery done?
Table 1 and 2 are containing Italian words and abbreviations are not explained under the table
Line 101 Rephrase, the word “bodies” is not proper jargon
“Additionally, there was swelling of the uterine and vaginal bodies, suggestive of estrus. “ So was she in oestrus during the visit or wasn’t she?
Line 103 : “Following confirmation, the dog underwent an exploratory laparoscopy to remove the suspected ovarian remnant, which was subsequently preserved in 4% formalin at room temperature until histological examination.” Was this the same day, week or one month later? Timing not clear. In the discussion the authors refer to corpora lutea and to the oestrous symptoms at the time of surgery.
Image 1 should be Figure 1 : A, B and C is indicated in red instead of white and barely visible in the figures.
Imagine 2 should be Figure 2: It is not indicated where the follicles are in B, also magnification not indicated and proper evaluation s not possible due to lack of detail in these figures.
Signs started 10 months before presentation. How long after OVH and how long before final surgery did she have her last oestrus?
Line 137 : “Nevertheless, in some cases, remnant of ovarian tissue might not produce sufficient estradiol levels capable to determine a feedback mechanism of pituitary gonadotropins [8] “ Even for someone who knows about this feedback mechanism, it is not clear what is meant here: do you mean that in these cases high or low levels of gonadotropins are identified ? It is indeed true that gonadotropin levels can also give additional information for diagnosis of ORS : Low level of LH in serum is present in ORS , which is high in neutered animals due to lack of negative feedback.
This is not at all explained in the paper, whereas the authors presume this knowledge to exist with the reader given their remark in the discussion see line 136- 137 :
Other interesting references that were not mentioned :
https://obgyn.onlinelibrary.wiley.com/doi/abs/10.1111/j.1447-0756.2012.02068.x
https://pubmed.ncbi.nlm.nih.gov/38237213/
https://pubmed.ncbi.nlm.nih.gov/31050839/
Comments on the Quality of English LanguageEnglish language should be revised
Author Response
General comments
This is a case report on an important topic, which is of interest to small animal vets. It is often indeed a problem if a bitch is suspected of ORS and has low AMH: vets are then in doubt of the diagnosis and need to know a way on how to proceed. Here CT scan is proposed but other solutions are also possible.
However the paper is not acceptable in its current form and should be rewritten completely. A few suggestions for improvement are written below and in the specific comments, but the paper should also be checked for English language and figures need profound editing. The timing of all events needs to be better described in a timeline of the history of the dog.
Dear reviewer, thank you for your comments which are useful in oreder to improve the manuscript. Every suggestion listed above was done in the text.
- In the introduction it should be explained better how diagnosis of ORS can be performed. There are different blood tests and other tests which can be performed for diagnosis of ORS – references to these method can be found in Manuals of dog reproduction or in papers such as cited here : “Currently, diagnostic methods such as medical history, oestrous symptoms, vaginal cytology (if in the follicular phase), hormone analysis (oestrogen and progesterone concentrations after GnRH stimulation test) and exploratory laparotomy are used to establish a diagnosis in ORS suspected dogs”(ref https://onlinelibrary.wiley.com/doi/10.1111/rda.13451)
This information has been added in the introduction part.
- Do not use the word sexual hormones. It is preferable to refer to steroid hormones (E2 and P4) and gonadotropins (LH & FSH).
This has been modified
- Indicate that just like when determining optimal timing of breeding or insemination we do not rely solely on progesterone, diagnosis of ORS cannot rely solely on AMH. It is the combination of all symptoms that leads us to the diagnosis.
This informatiomn has been addeed at the end of the discussion section.
The paper needs a thorough rereview before it can be accepted for publication.
Specific comments
Simple summary needs to be revised for language
English has been revised and edited.
Abstract needs to be revised too
English has been revised and edited.
line 27-30 Please remove one of the following sentences since they give the same information :
“Following advanced diagnostics and laparoscopy, histological examination confirmed the presence of ovarian tissue in the dog’s abdominal cavity. In this clinical case, in addition to hormonal tests, a diagnosis was made using CT scans and laparoscopy, confirming the presence of ovarian tissue in the dog’s abdominal cavity”
A sentence has been removed
Line 37 : : why is ORS a non-pathological condition? Please explain better or remove non-pathological
It has been removed
Line 45 : please rephrase : it is not sexual hormones that induce keratinization (or better cornification) but estrogens . Progesterone is also “a sexual hormone” (not sure if this is proper terminology) but does not induce cornification in the vagina.
It has been clarified.
Line 72: What is meant by a partial ovariohysterectomy? It is also not clear how the timing of OVH relates to the serosanguinous blood loss. Was the blood loss ten months before the OVH and the OVH 10 months before the current visit (so blood loss 20 months ago?), or did the blood loss and OVH occur at the same time (meaning that she should have had another estrous cycle about 4 to 2 months ago (6-8 months interoestrous interval in most dogs). Was the dog spayed during oestrus or 3 months after oestrus as adviced in textbooks? At what time of the cycle was the final curative surgery done?
Partial ovariohysterectomy refers to a procedure that removes the cervix incompletely. For the timing of the clinic, we rewrote the sentence like this: The owner reported that Callie had undergone a partial ovariohysterectomy about 10 months prior to the visit to halt her reproductive activity, and after a few days the vulvar discharge started.
Table 1 and 2 are containing Italian words and abbreviations are not explained under the table. Done.
Line 101 Rephrase, the word “bodies” is not proper jargon
The term has been removed
“Additionally, there was swelling of the uterine and vaginal bodies, suggestive of estrus. “ So was she in oestrus during the visit or wasn’t she?
The sentence has been modified.
Line 103 : “Following confirmation, the dog underwent an exploratory laparoscopy to remove the suspected ovarian remnant, which was subsequently preserved in 4% formalin at room temperature until histological examination.” Was this the same day, week or one month later? Timing not clear.
This has been clarified.
In the discussion the authors refer to corpora lutea and to the oestrous symptoms at the time of surgery.
The information has been clarified. The unique symptom observed was vaginal discharge that needs to be taken to differential. The female was not attracting males neither having vaginal cells as keratinized because of estradiol impregnation. The reason for this discharge is still unknown to the authors.
Image 1 should be Figure 1 : A, B and C is indicated in red instead of white and barely visible in the figures.
The elements have been modified
Imagine 2 should be Figure 2: It is not indicated where the follicles are in B, also magnification not indicated and proper evaluation is not possible due to lack of detail in these figures.
Done
Signs started 10 months before presentation. How long after OVH and how long before final surgery did she have her last oestrus?
The owners reported that the bloody discharge was present more or less intense throughout the 10 months after surgery.
Line 137 : “Nevertheless, in some cases, remnant of ovarian tissue might not produce sufficient estradiol levels capable to determine a feedback mechanism of pituitary gonadotropins [8] “ Even for someone who knows about this feedback mechanism, it is not clear what is meant here: do you mean that in these cases high or low levels of gonadotropins are identified ? It is indeed true that gonadotropin levels can also give additional information for diagnosis of ORS : Low level of LH in serum is present in ORS , which is high in neutered animals due to lack of negative feedback.
This is not at all explained in the paper, whereas the authors presume this knowledge to exist with the reader given their remark in the discussion see line 136- 137 :
All the previous information has been clarified
Other interesting references that were not mentioned :
https://obgyn.onlinelibrary.wiley.com/doi/abs/10.1111/j.1447-0756.2012.02068.x
https://pubmed.ncbi.nlm.nih.gov/38237213/
https://pubmed.ncbi.nlm.nih.gov/31050839/
Done.
Comments on the Quality of English Language
English language should be revised
English has been checked by a proficiency level person
Reviewer 2 Report
Comments and Suggestions for Authors
First of all I have been reviewed the paper entitled:Unexpected basal Anti-Müllerian Hormone concentrations in a 2 6-year-old bitch presenting an ovarian remnant. The paper is very important and good with interesting idea but need more information and details
Abstract is the AMH measured is specific for German shepherd dog? Please clarify
The last paragraph in the abstract is very bad written this part is very important as it's the conclusion part must be informative and clear
Keywords must be informative and not contain the same words
Introduction
Line 44-48 need more references with more information and details about the clinical signs
Line 54-56 please revise this sentence contains error in Grammer
Line 69 what about inclusion and exclusion criteria?
What about details of stain ?
What about anti muller an hormones detection? I mean what is pre analysis and analysis phase in addition to Post analysis phase
In tables author must pay attention to write all the details provided in table title how come I read table with general title
First paragraph of discussion must be rewritten again
Line 160-165 need rephrasing again
Line 170 please add more references that were similar to your data
Conclusion is ok
Author Response
First of all I have been reviewed the paper entitled:Unexpected basal Anti-Müllerian Hormone concentrations in a 2 6-year-old bitch presenting an ovarian remnant. The paper is very important and good with interesting idea but need more information and details
Dear reviewer, thank you for your comments which are useful in oreder to improve the manuscript. Every suggestion listed above was done in the text.
Abstract is the AMH measured is specific for German shepherd dog? Please clarify
There are no available kits for AMH inn different breeds. To date the most specific kit for measuring AMH in dogs is the kit that we used. The used kit is described in the case desciption.
The last paragraph in the abstract is very bad written this part is very important as it's the conclusion part must be informative and clear
The whole abstract has been rewritten
Keywords must be informative and not contain the same words
Introduction
Line 44-48 need more references with more information and details about the clinical signs
Done
Line 54-56 please revise this sentence contains error in Grammer
It has been rewritten
Line 69 what about inclusion and exclusion criteria?
There was not an inclusion criteria as the manuscript is a case report and not a scientific study.
What about details of stain ?
Details of the stain were already in the manuscript as: (Dif-stain, Titolchimica S.r.l., Pontecchio Polesine, Rovigo, Italy)
What about anti muller an hormones detection? I mean what is pre analysis and analysis phase in addition to Post analysis phase
Anti Müllerian hormone can be detected with just one analysis showing some non-significant variations between the phases during the cycle as it is decribed in the introduction section
In tables author must pay attention to write all the details provided in table title how come I read table with general title.
Done
First paragraph of discussion must be rewritten again
It has been rewritten
Line 160-165 need rephrasing again
It has been rephrased and english was improved
Line 170 please add more references that were similar to your data
Done
Conclusion is ok
Thanks

Round 2
Reviewer 1 Report
Comments and Suggestions for Authors
The manuscript has substantially improved but still some parts need revision before the paper can be published.
Line 58 The presence of sexual steroids hormones in the bloodstream- replace by The presence of gonadal steroid hormones
Table 1 : Linfocytes should be lymphocytes
The tables give a lot of data but only a few relevant changes are present. But the authors state - The results of the -hematological and biochemical examinations showed no values that deviated from the normal reference ranges (Table 1, 2).
Is this really so? Neutrophils and CRP is elevated. Could this be an indication for the swollen uterus? Stump pyometra? It would also explain the blood loss. but on line 141 the authors say the swollen uterus is because of estrus whereas there are no keratinized cells in the vagina (line 107)
ALT is also higher than normal.
I suggest to discuss these values or otherwise remove the tables. There are still spelling mistakes in both tables (Italian spelling) , please check carefully.
Line 180 Same sentence is repeated twice : "In the reported case, histological examination confirmed aging corpora lutea, suggesting functional ovarian tissue was present. On the other hand, histological examination confirmed the presence of aging corpora lutea"
Comments on the Quality of English LanguageStill needs to be improved
Author Response
The manuscript has substantially improved but still some parts need revision before the paper can be published.
Dear reviewer, first of all thank you for your appreciation of my work and also for your criticisms, which areuseful for improving it.
Line 58 The presence of sexual steroids hormones in the bloodstream- replace by The presence of gonadal steroid hormones
Done
Table 1 : Linfocytes should be lymphocytes
Done
The tables give a lot of data but only a few relevant changes are present. But the authors state - The results of the -hematological and biochemical examinations showed no values that deviated from the normal reference ranges (Table 1, 2). Is this really so? Neutrophils and CRP is elevated. Could this be an indication for the swollen uterus? Stump pyometra? It would also explain the blood loss. but on line 141 the authors say the swollen uterus is because of estrus whereas there are no keratinized cells in the vagina (line 107). ALT is also higher than normal. I suggest to discuss these values or otherwise remove the tables. There are still spelling mistakes in both tables (Italian spelling) , please check carefully.
Spelling errors have been corrected. Sentences have been added to the text to justify parameters that are slightly out of range, but still not pathology-specific. Corrections have been highlighted in yellow to make it easier to proofread the text.
Line 180 Same sentence is repeated twice : "In the reported case, histological examination confirmed aging corpora lutea, suggesting functional ovarian tissue was present. On the other hand, histological examination confirmed the presence of aging corpora lutea"
A repeated phrase has been removed.
